# Increased inflammatory signature in myeloid cells of non-small cell lung cancer patients with high clonal hematopoiesis burden

**Hyungtai Sim[1†], Hyun Jung Park[2,3†], Geun-Ho Park[4,5†], Yeon Jeong Kim[2], Woong-Yang Park[2], Se-Hoon Lee[4,5*‡], Murim Choi[1*‡]**

[1]Department of Biomedical Sciences, Seoul National University College of Medicine, Seoul, Republic of Korea; [2]Samsung Genome Institute, Samsung Medical Center, Seoul, Republic of Korea; [3]Research Institute for Veterinary Science, College of Veterinary Medicine, Seoul National University, Seoul, Republic of Korea; [4]Division of Hematology-Oncology, Department of Medicine, Samsung Medical Center, Sungkyunkwan University School of Medicine, Seoul, Republic of Korea, Seoul, Republic of Korea; [5]Department of Health Sciences and Technology, Samsung Advanced Institute for Health Science & Technology (SAIHST), Sungkyunkwan University, Seoul, Republic of Korea

*For correspondence:
sehoon.lee119@gmail.com (S-HL);
murimchoi@snu.ac.kr (MC)

†These authors contributed equally to this work
‡These authors also contributed equally to this work

## eLife Assessment

This **valuable** article represents a significant body of work that addresses some novel aspects of the biology of lung cancer, the overall influence of CHIP and its impacts on responses to therapy. While a high clonal hematopoiesis (CHIP) burden was previously linked with an inflammatory phenotype in other disease settings, the authors demonstrate with **solid** evidence that this is also true for lung cancer. CHIP is complex and more data will be required to substantiate more evidence with regard perhaps to specific mutations in certain situations and how this might influence therapy choices.

**Abstract** Clonal hematopoiesis of indeterminate potential (CHIP) allows estimation of clonal dynamics and documentation of somatic mutations in the hematopoietic system. Recent studies utilizing large cohorts of the general population and patients have revealed significant associations of CHIP burden with age and disease status, including in cancer and chronic diseases. An increasing number of cancer patients are treated with immune checkpoint inhibitors (ICIs), but the association of ICI response in non-small cell lung cancer (NSCLC) patients with CHIP burden remains to be determined. We collected blood samples from 100 metastatic NSCLC patients before and after ICI for high-depth sequencing of the CHIP panel and 63 samples for blood single-cell RNA sequencing. Whole exome sequencing was performed in an independent replication cohort of 180 patients. The impact of CHIP status on the immunotherapy response was not significant. However, metastatic lung cancer patients showed higher CHIP prevalence (44/100 for patients vs. 5/42 for controls; p = 0.01). In addition, lung squamous cell carcinoma (LUSC) patients showed increased burden of larger clones compared to lung adenocarcinoma (LUAD) patients (8/43 for LUSC vs. 2/50 for LUAD; p = 0.04). Furthermore, single-cell RNA-seq analysis of the matched patients showed significant enrichment of inflammatory pathways mediated by NF-κB in myeloid clusters of the severe CHIP group. Our findings suggest minimal involvement of CHIP mutation and clonal dynamics during immunotherapy but a possible role of CHIP as an indicator of immunologic response in NSCLC patients.

## Introduction

Recent studies have revealed that somatic mutations detected in hematopoietic malignancies are also commonly observed with clonal expansion in the individuals with no hematological conditions (*Jaiswal et al., 2014*). Among such individuals, asymptomatic cases with a variant allele frequency (VAF) of ≥2% are defined as clonal hematopoiesis of indeterminate potential (CHIP) (*Jaiswal and Ebert, 2019*). In general, CHIP is strongly associated with aging; and mutagenic anticancer therapies and smoking can also induce clonal expansion in specific genes (*Jaiswal et al., 2014*; *Hsu et al., 2018*). Longitudinal monitoring of CHIP mutations offers important clues into the clonal dynamics of blood (*Uddin et al., 2022*).

Hematopoietic stem and progenitor cells (HSPCs) carrying CHIP mutations serve as a major source of expanded clones, and even a few cells are sufficient to influence clonal homeostasis (*Welch et al., 2012*). Representative CHIP mutation-carrying genes, such as *DNMT3A*, *ASXL1*, and *TET2*, impact the myeloid population, including monocytes and granulocytes, and have been linked to hematologic malignancies such as myelodysplastic syndrome and acute myeloid leukemia through inflammation mediated by the NF-κB and NLRP3 inflammasome (*Muto et al., 2020*; *Cai et al., 2018*; *Abplanalp et al., 2021*). The prevalence of clonal hematopoiesis (CH) in cancer patients is also significantly higher than that in healthy populations, and high CH burden has been implicated in a negative effect on overall survival in multiple cancer types (*Coombs et al., 2017*; *Bolton et al., 2020*). CHIP is also associated with diseases beyond hematologic malignancies, such as atherosclerosis, type 2 diabetes, and chronic obstructive pulmonary disease (*Miller et al., 2022*; *Tobias et al., 2023*; *Jaiswal et al., 2017*). Multiple studies, including both population and pan-cancer cohorts, have explored the correlation of CHIP with solid tumors (*Coombs et al., 2017*; *Bolton et al., 2020*; *Xie et al., 2014*; *Stacey et al., 2023*). However, establishing a causal relationship has proven challenging due to confounding factors such as history of smoking and anticancer therapy, both of which act as risk factors for CHIP (*Bolton et al., 2020*).

Non-small cell lung cancer (NSCLC) stands out globally as the solid tumor with the highest mortality burden (*Leiter et al., 2023*). The difficulty in early detection of NSCLC significantly contributes to its mortality rate being high compared to its incidence rate. Recent studies examining CHIP within lung cancer cohorts and general population indicate a possible association of lung cancer risk with CH (*Stacey et al., 2023*; *Hong et al., 2022*; *Tian et al., 2023*). As mentioned above, factors such as a patient's history of anticancer treatments and smoking serve as common risk factors for CH; however, the interplay between lung cancer and CHIP is not yet fully understood. Immune checkpoint inhibitors (ICIs) targeting PD-(L)1 and CTLA-4 have revolutionized treatment of advanced NSCLC, both as monotherapies and in combination with chemotherapy (*Reck et al., 2021*). However, response rates to ICI treatments remain relatively low. The predictive ability of markers like PD-1 expression and tumor mutational burden is insufficient, requiring the identification of additional markers.

Recent studies suggest a marginal effect of ICI on the clonal dynamics of CHIP (*Miller et al., 2021*; *Miller et al., 2020*), but the current understanding of the role of CHIP in ICI-treated patients is not comprehensive. Also, only limited analysis has been performed concerning the impact of CHIP on cancer patients' blood. To understand the interplay between ICI and CHIP in NSCLC patients, we conducted CHIP-targeted panel sequencing and single-cell RNA sequencing (scRNA-seq) on blood samples collected before and 1–3 weeks after ICI administration. Our results indicate a marginal interaction between ICI and CHIP, but also validate that high CHIP burden manifests an inflammatory phenotype in metastatic NSCLC patients' myeloid cells.

## Results

### CHIP profiles in metastatic NSCLC patients treated with ICI

To investigate the interplay between ICI treatment and CHIP in NSCLC, we collected blood samples before and after ICI treatment from 100 metastatic NSCLC patients (*Figure 1*, *Supplementary file 1a*). These samples were utilized for a CHIP-targeted sequencing panel (median depth ~600×) and scRNA-seq (*Figure 1A* and Methods). At the baseline, we identified 67 CHIP mutations in 100 patients, 26 of them in known putative driver genes of hematopoietic cancer (CHIP-PD). After treatment, 68 CHIP and 32 CHIP-PD mutations were found in 91 patients who passed quality control assessment (*Figure 1A*, *Figure 1—figure supplements 1–3*, and *Supplementary file 1c*).

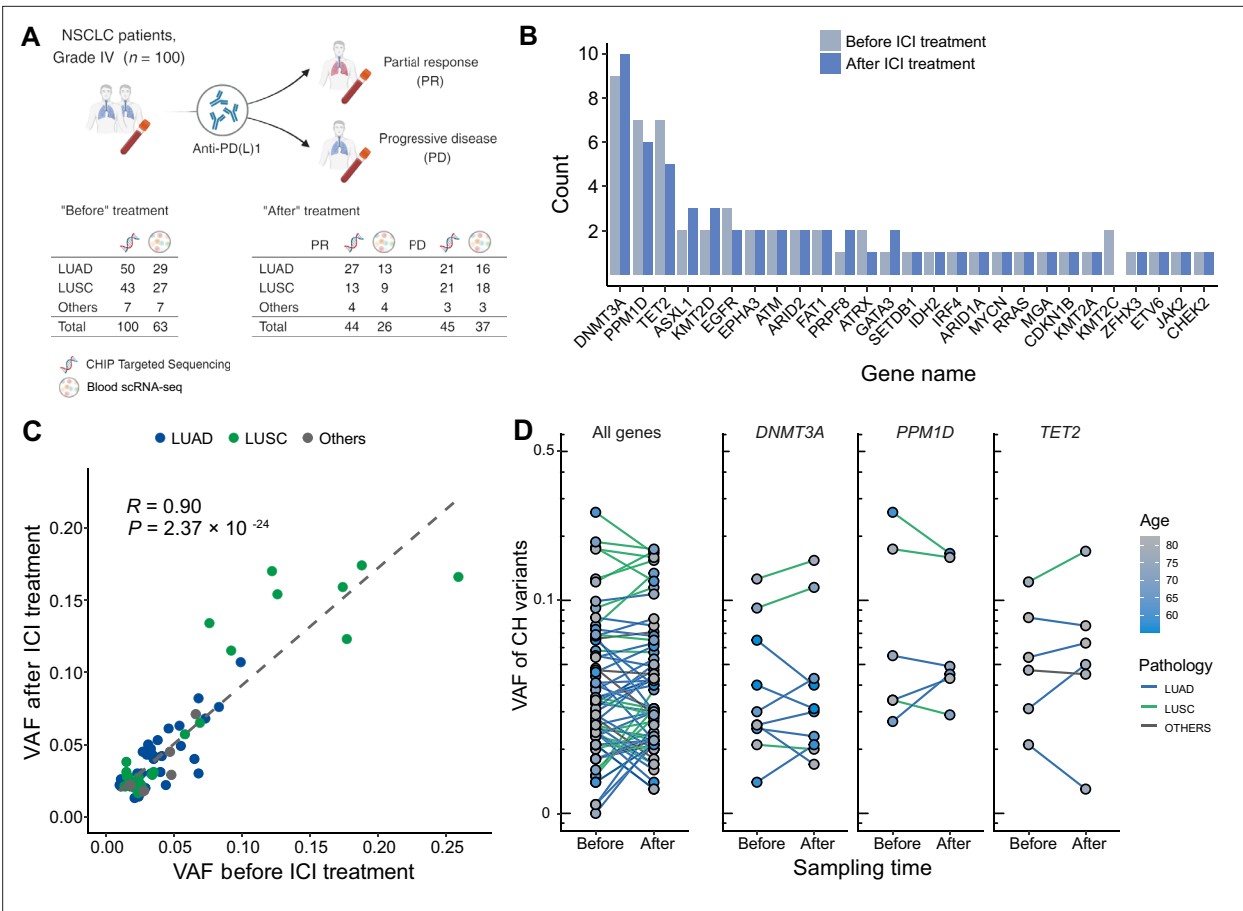

**Figure 1.** Profile of CH in non-small cell lung cancer (NSCLC) patients treated with immune checkpoint inhibitor (ICI). (**A**) Overall study design. The numbers indicate sample counts in each group. (**B**) Frequency of CH variant detection in each gene in NSCLC samples before (light blue) and after (blue) ICI treatment. (**C**) Comparison of variant allele frequency before and after ICI treatment, divided by pathology type (lung adenocarcinoma [LUAD], lung squamous cell carcinoma [LUSC], other). (**D**) Effect of ICI treatment on the clonal landscape of all genes (left) and frequently detected CH genes.

The online version of this article includes the following figure supplement(s) for figure 1:

**Figure supplement 1.** Variant calling and filtering scheme.

**Figure supplement 2.** Depth and annotation distributions from variants filtering scheme.

**Figure supplement 3.** Clonal hematopoiesis of indeterminate potential (CHIP) profiles from discovery cohort with panel sequencing (sample $n$ = 100/91 for before treatment and after treatment).

Among the CHIP genes, *DNMT3A*, *PPM1D*, and *TET2* were the most frequently mutated (*Figure 1B*). *PPM1D* truncating mutations were also common, and have previously demonstrated association with patients' smoking habits and possible history with chemotherapy (*Hsu et al., 2018*; *Coombs et al., 2017*). Variants involving the *DNMT3A* p.Arg882 residue and *TP53*, which are frequent in CHIP, were not prevalent in our dataset, although our panel extensively covered these regions and these loci were manually inspected (data not shown).

Overall, we observed ICI treatment to effect minimal changes in CHIP burden (*Figure 1B*). Also, VAF was highly correlated across all traced mutations and not dependent on clinical responses (*Figure 1C, D*). However, we noted increased number of CHIP mutations with VAF >10% in patients with lung squamous cell carcinoma (LUSC). While the specific mutated genes differed between patients with different histology or responses, mutated gene burden did not significantly differ between groups.

## Prevalence of high CHIP burden showed bias to LUSC patients

Next, we investigated the association of CHIP status with clinical parameters in the cohort (*Figure 2*). CHIP prevalence was significantly higher in the lung cancer cohort compared to the control group (controls 5/42 vs. patients 44/100, age-adjusted logistic regression p = 0.01; *Figure 2A*, *Supplementary*

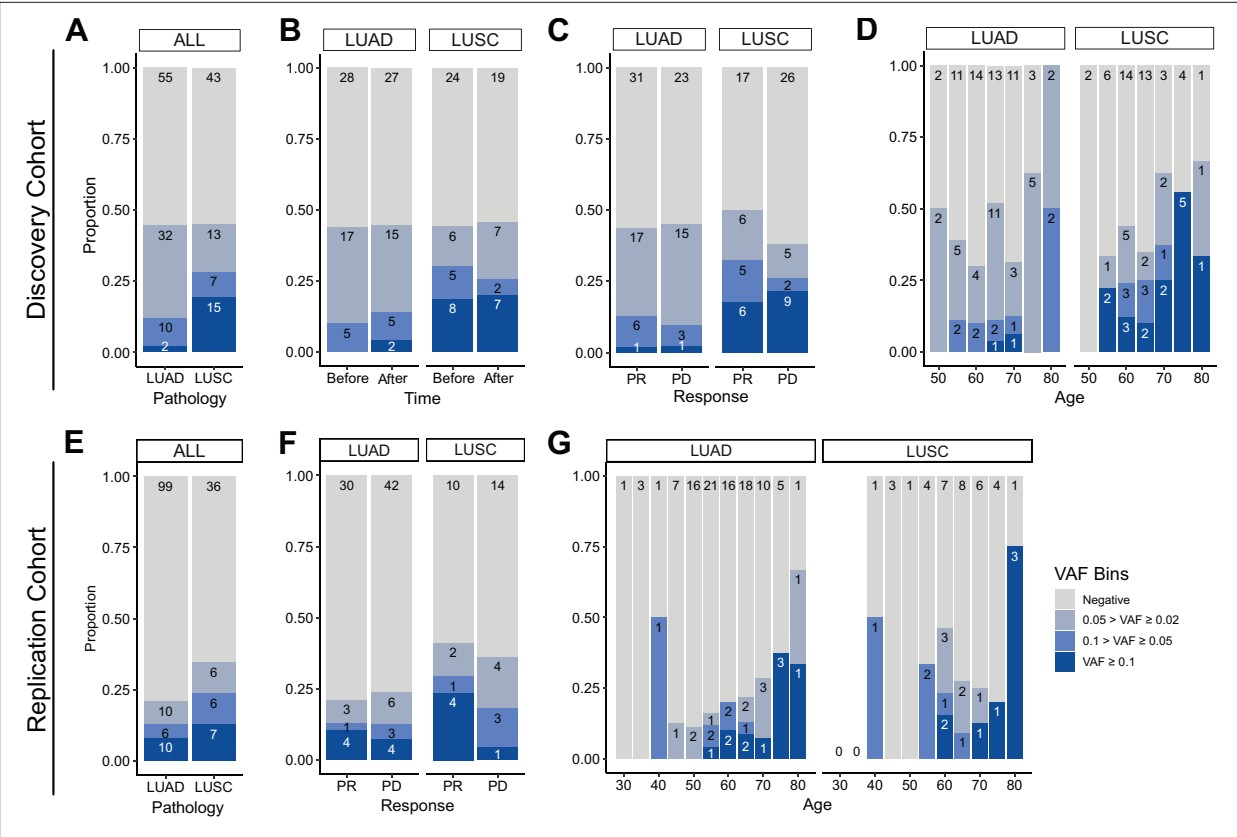

**Figure 2.** Prevalence of CH in relation to various parameters. (**A–D**) Discovery cohort (*n* = 100), (**E–G**) replication cohort (*n* = 180; see Methods for description). (**A, E**) CH prevalence by pathology (lung squamous cell carcinoma [LUSC] and lung adenocarcinoma [LUAD]). (**B**) Effect of immune checkpoint inhibitor (ICI) treatment on CH. (**C, F**) Effect of post-ICI prognosis to ICI on CH. (**D, G**) Age distribution of the cohort, stratified by clonal hematopoiesis of indeterminate potential (CHIP) allele frequency and pathology status.

The online version of this article includes the following figure supplement(s) for figure 2:

**Figure supplement 1.** Clonal hematopoiesis of indeterminate potential (CHIP) profiles, by stratified smoking status.

**Figure supplement 2.** Relationship between clonal hematopoiesis of indeterminate potential (CHIP) variant counts and clinical parameters.

**Figure supplement 3.** Clonal hematopoiesis of indeterminate potential (CHIP) profiles from replicative cohort with peripheral blood mononuclear cell (PBMC) whole exome sequencing (WES) (*n* = 180).

*file 1a and b*). However, the number of CHIP-positive patients did not differ before and after ICI treatment, nor in relation to treatment response (*Figure 2B, C*). As previously reported, CHIP burden is heavily dependent on age (*Figure 2D*). Smoking history may also impact CHIP prevalence, but we could not find a significant contribution of smoking, as almost all NSCLC patients in our study have a smoking history (CHIP prevalence of non-smoker 2/9 vs. smoker 42/91; *Figure 2—figure supplement 1*).

Subsequently, we examined clonal size in the cohort, as that has been reported as a risk factor of clinical outcome (*Kessler et al., 2022*). Taking CHIP VAF as an indicator of clonal size, we observed that high CHIP burden was prevalent in LUSC, while multiple concurrent CHIP mutations were common in lung adenocarcinoma (LUAD) patients (*Figure 2C* and *Figure 2—figure supplement 2*). To determine if this finding would reproduce in an independent cohort, we performed CHIP mutation calling using whole exome sequencing (WES) data of median ~80× depth from 180 additional peripheral blood mononuclear cell (PBMC) samples from an independent cohort of NSCLC patients (*Figure 2C–E*, *Figure 2—figure supplement 3*, *Supplementary file 1a, c, d* and Methods). The results showed similar patterns between the two cohorts, but the replication cohort alone was not powerful enough to achieve significance for difference in clonal sizes based on pathology (*Figure 2E–G*).

## High CHIP burden causes single-cell transcriptomic changes

Next, to test whether the presence of CHIP and ICI prognosis affects blood cell transcriptomes, we analyzed PBMC scRNA-seq data from 63 samples of the discovery cohort. After quality control processes, we retrieved 468,596 cells in a total of 26 clusters defined by systemic integration and clustering using the Louvain algorithm, then validated using Azimuth (*Figure 3A*, *Figure 3—figure supplement 1A, B*, *Supplementary file 1e*, and Methods) (*Hao et al., 2021*). Cell composition analysis showed an increased proportion of natural killer (NK) cells and a decreased CD4 TEM population after ICI administration, but high variability between samples precluded systematic differences by CHIP status (*Figure 3—figure supplement 1C*). Subsequently, differentially expressed genes (DEG) analysis using Wilcoxon's rank-sum test according to CHIP VAF bins in each cluster (see Methods) highlighted genes important in immune response and transcriptional activation (*Figure 3C*, *Figure 3—figure supplement 2*, *Supplementary file 1f*). To further implicate these DEGs functionally, a gene set enrichment test was performed using the area under the curve (AUC) statistics from DEGs and the hallmark pathway gene sets in msigDB (*Liberzon et al., 2015*). The results indicated activation of the tumor necrosis factor (TNF) signaling pathway via *NFKB* in the high CH burden group for most cell lineages, including both classical dendritic cells (DCs) and NK cells (*Figure 3D, E*, *Figure 3—figure supplement 3*, and *Supplementary file 1g*; see Methods). To understand possible interactions between ICI treatment response and CHIP burden, we grouped samples by CHIP burden and ICI response (*Figure 3—figure supplement 4*). However, the interactions mostly depended on CH status only, and not on response to ICI.

## Implication of high CHIP burden in inflammatory signatures of myeloid cells

We performed a gene network analysis to comprehensively analyze the increased expression of inflammatory signaling genes in patients with higher CHIP burden. First, we used hdWGCNA to construct co-expression networks for genes in myeloid cells, which yielded a black module that exhibited myeloid specificity and represented most of the signaling-related DEGs identified via gene set enrichment analysis (GSEA) (*Figure 3—figure supplement 5* and *Supplementary file 1h*; *Morabito et al., 2023*). We determined gene ontology term enrichment in this module and found the inflammatory and transcriptional activation pathways observed in DEGs to be clearly identified (*Figure 3—figure supplement 5C*). Additionally, further GSEA on the myeloid-specific module showed its function to resemble that of the overall set of DEGs (*Figure 3—figure supplement 5D*).

In addition to weighted correlation network analysis (WGCNA), gene regulatory network (GRN) analysis using SCENIC (*Van de Sande et al., 2020*; *Figure 3—figure supplement 6*) indicated that the regulatory activity of transcription factors in the AP-1 and NF-κB pathways was particularly notable in the myeloid lineage of high-burden patients (*Figure 3—figure supplement 6A, B*). Both the NF-κB regulon and the ATF3 regulon, which represents the AP-1 pathway, were activated in the myeloid cluster of high-burden patients (*Figure 3—figure supplement 6C–E*).

Recent studies have shown that expansion of HSPC-derived monocytes is the cause of CHIP, and representative CHIP genes, including *DNMT3A* and *TET2*, exert immune-regulatory effects through the NLRP3 inflammasome (*Welch et al., 2012*; *Abplanalp et al., 2021*; *Sano et al., 2018*). Our network analysis also points to this pathway (*Figure 3—figure supplement 6*). Notably, a VAF of 10% in heterozygote mutations corresponds to approximately 20% of cells; thus, the presence of such mutations in HSPC-derived monocytes of high-burden patients would have resulted in an overall transcriptional profile change for blood monocytes.

In addition to conducting gene network analysis, we explored the effect of the inflammatory signature on cell–cell interaction using CellPhoneDB (*Figure 3—figure supplement 7*; *Efremova et al., 2020*). The number of significant cell–cell interactions did not differ significantly between high-burden CHIP samples and negative samples, although the number of cells in each group seemed likely to have an impact (*Figure 3—figure supplement 7A, B*). However, when the interactions in the TNF and IL-1B pathways were analyzed separately, monocytes and DCs in high-burden cases were activated as receptors for TNF and Lymphotoxin A (*Figure 3—figure supplement 7C*). Collectively, our results suggest a sensitized response of monocytes to inflammation in lung cancer patients with high CHIP burden.

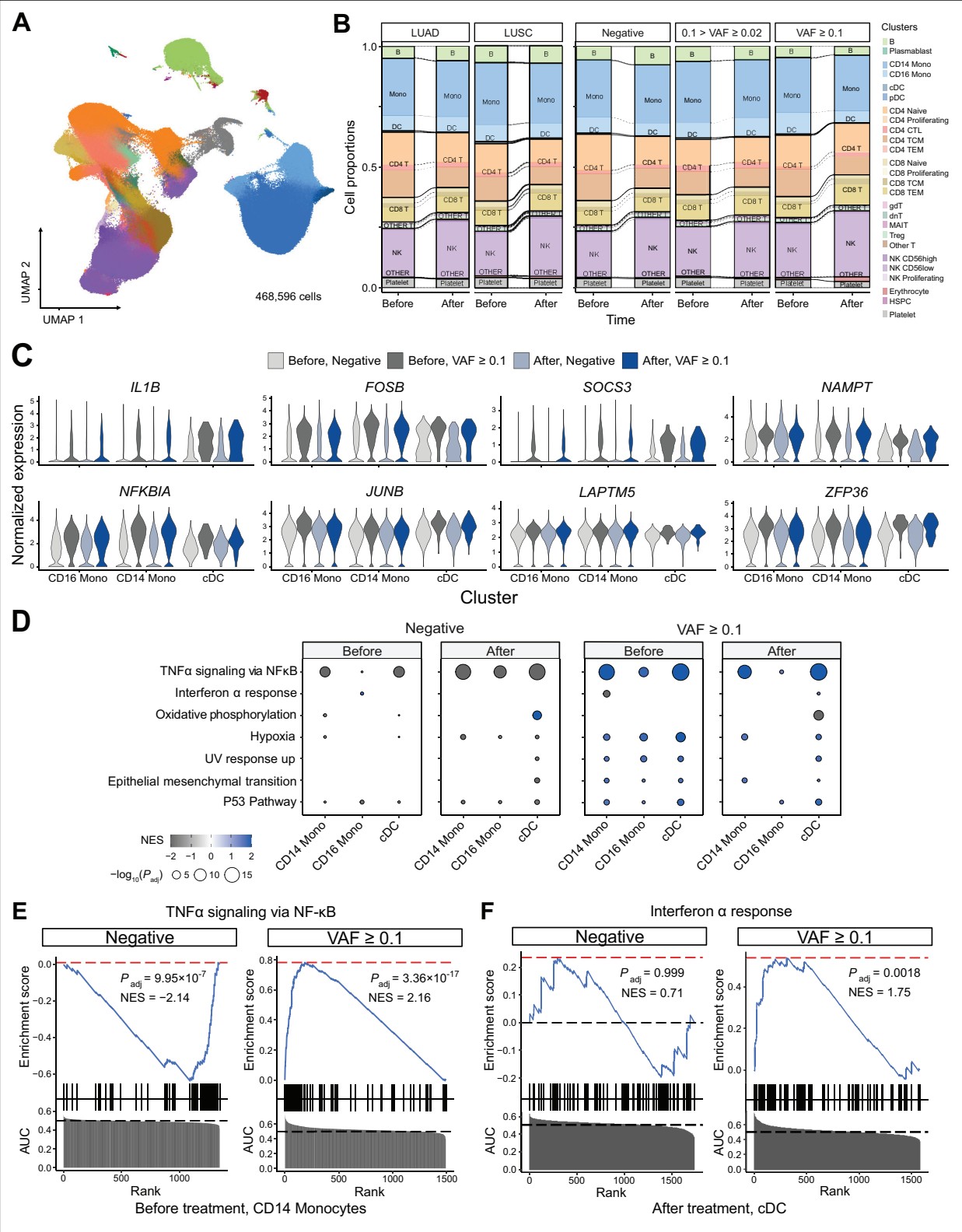

**Figure 3.** Single-cell RNA sequencing (scRNA-seq) analysis detected myeloid-specific inflammatory signatures in patients with high CH burden. (**A**) Uniform manifold approximation and projection (UMAP) plot of scRNA-seq data from the discovery cohort ($n$ = 63). (**B**) Effect of pathology (left) and variant allele frequency (VAF) (right) on cell composition. (**C**) Expression of selected genes in the NF-κB pathway from the scRNA-seq data. (**D**) Gene set enrichment analysis (GSEA) of DEGs from myeloid populations. Color represents normalized effect score (NES), and dot size represents adjusted p-values. (**E**) and (**F**) GSEA plots of selected pathways from (**C**).

*Figure 3 continued on next page*

*Figure 3 continued*

The online version of this article includes the following figure supplement(s) for figure 3:

**Figure supplement 1.** Single-cell RNA sequencing (scRNA-seq) profile of immune checkpoint inhibitor (ICI)-treated, non-small cell lung cancer (NSCLC) patients.

**Figure supplement 2.** Gene expression of selected pathways for (**A**) IL-10 and (**B**) IL-1B.

**Figure supplement 3.** Gene set enrichment analysis (GSEA) results from all annotated clusters.

**Figure supplement 4.** Gene set enrichment analysis (GSEA) results and representative gene expression patterns, associated with immune checkpoint inhibitor (ICI) response and clonal hematopoiesis of indeterminate potential (CHIP) variant allele frequency (VAF) bin.

**Figure supplement 5.** Single-cell RNA sequencing (scRNA-seq) weighted correlation network analysis (WGCNA) using hdWGCNA.

**Figure supplement 6.** SCENIC analysis for gene regulatory networks (GRNs) on single-cell RNA sequencing (scRNA-seq).

**Figure supplement 7.** Cell–cell interaction analysis using CellphoneDB.

**Figure supplement 8.** Survival plot from this cohort.

**Figure supplement 9.** Bar plot of the top shared gene set enrichment analysis (GSEA) leading edge genes, using significantly enriched (adjusted p < 0.05 for each cluster).

## Effect of CHIP on clinical outcome

Finally, to understand whether clonal hematopoiesis status can affect the clinical outcomes of lung cancer patients undergoing ICI treatment, we conducted a survival analysis (*Figure 3—figure supplement 8*). Our results showed the progression-free survival (PFS) of patients after ICI treatment to be reduced in the high-burden CHIP group (825 vs. 404, p = 0.056), with no difference in ICI response rates between the negative and high-burden CHIP groups. Previous studies have already observed a negative effect on life expectancy in CHIP patients with solid tumors. Therefore, it is possible that these results will be validated by analyzing additional samples.

## Discussion

To address the influence of CH in lung cancer patients with differential prognosis following ICI, we sought to determine the following: (1) CH profiles from metastatic NSCLC patients, (2) changes in PBMC gene expression by scRNA-seq, (3) relationship of CH burden and transcriptome profile, and (4) potential implication for the clinical outcomes. Here, we report that CH prevalence was significantly higher in NSCLC patients and higher burden of mutations with VAF of greater than 10% was frequently observed in LUSC. Using scRNA-seq analysis, we found myeloid cells in NSCLC patients with high CH burden to exhibit an altered inflammatory milieu via increased NF-κB pathway expression. Our results provide possible explanations for heterogeneity in response to immunotherapy and suggest immune response to be altered due to CHIP in cancer patients.

Recent studies have employed mouse models and single-cell transcriptomics to understand the roles of specific mutations in CH and provided evidence of their implications for inflammatory response in the myeloid lineage (*Nam et al., 2022*; *Fuster et al., 2017*). In particular, a study linking COVID-19 and CH showed an increased inflammatory response from monocytes in severe COVID-19 patients with CH, mediated via interferon gamma (IFN-γ) and TNF-α signaling signatures (*Choi et al., 2022*). Likewise, we observed that myeloid cells from NSCLC patients with CH tend to harbor increased inflammatory response, and this response is mediated by NF-κB pathway signaling (*Figure 3*).

Our GSEA results specifically indicated the enrichment of TNF signaling and hypoxia pathways in most clusters of patients with severe CH (*Figure 3—figure supplement 3*). The leading edge genes from GSEA results showed core genes such as *FOS*, *DUSP1*, *JUN*, and *PPP1R15A*, which are known to play critical roles in the inflammatory phenotypes of immune cells, were shared between the TNF signaling and hypoxia pathways in all significant clusters (*Figure 3—figure supplement 9*). Furthermore, GRN analysis using SCENIC indicated a higher enrichment of inflammatory signatures in myeloid lineages (*Figure 3—figure supplement 6*), highlighting the pronounced inflammatory phenotype of CH clones in these cell lineages. To gain insight into the functional implications of myeloid-driven inflammatory gene pathways within CH in NSCLC patients, we conducted gene network analyses focusing on the myeloid lineage. The 'black module' from the WGCNA analysis (*Figure 3—figure supplement 5*) and the NFKB1 and ATF3 GRN from SCENIC (*Figure 3—figure*

*supplement 6*) revealed activation and adaptation of myeloid-specific inflammatory pathways (*Morabito et al., 2023*; *Van de Sande et al., 2020*). Additionally, cell–cell interaction analysis using cellphoneDB unveiled notable interactions between myeloid cells and other lineages mediated through the TNF pathway (*Figure 3—figure supplement 7 Efremova et al., 2020*).

As ICIs modulate pathways involved in immune response rather than cell proliferation, it is reasonable that ICIs do not exert a direct influence on CH development (*Miller et al., 2020*). Also, our results did not support significant difference of ICI outcome based on CH burden (*Figure 2*). According to our power analysis, a minimum of 100 samples per group with CHIP is required to achieve a statistical power of 0.8 from pair-wise comparisons between each gene and variant (Methods for details). However, our findings do suggest that elevated inflammatory activity in the bloodstream may impact long-term patient survival pre- and post-ICI administration (*Figure 3—figure supplement 8*). Therefore, a future study with a larger sample size may yet clarify a causal relationship of CH with ICI outcome. Notably, involvement of the IL-1 pathway in the aggressiveness and metastasis of lung cancer has been reported in experimental studies (*Li et al., 2020*; *Zhang and Veeramachaneni, 2022*), and inhibition of IL-1β has proven effective in reducing lung cancer-related mortality, as shown in the CANTOS trial (*Ridker et al., 2017*). Indeed, our analysis indicated a modest decrease in PFS in the ICI-responder group with high CH patients (*Figure 3—figure supplement 8*, p = 0.056). Our study design is potentially limited by the relatively short observation period for defining ICI response. Indeed, it is worth noting that more potent factors, such as smoking and anticancer therapies, may have a more substantial impact than immunotherapy (*Bolton et al., 2020*). Consistent with this postulation, our analyses indicated minimal changes in in CH-related signatures before and after ICI treatment (*Figure 3*, *Figure 3—figure supplement 1, 3, and 4*).

Also, the distinct characteristics of our cohort can be confounders for our results. Compared to control patients, our cohort was biased to slightly older ages, a higher prevalence of smoking habits, and with a higher proportion of males (mean age: 64.1 vs. 58.9; current smokers: 37/100 vs. 11/42; male/female: 91/9 vs. 18/24; *Supplementary file 1a, c*). However, previous studies have reported similar prevalence rates of clonal hematopoiesis in NSCLC patients, aligned with our findings (*Coombs et al., 2017*). Moreover, our most prevalent CH mutations, including *DNMT3A*, *TET2*, and *PPM1D*, were marginally affected by smoking, and this trend has been consistently observed in both healthy populations and NSCLC patients (*Bolton et al., 2020*; *Levin et al., 2022*).

Here, we have illustrated increased inflammatory activation in myeloid cells of NSCLC patients with high CH burden. Our scRNA-seq analysis has revealed that NF-κB signaling mediated by the IL-1β and the TNF superfamilies may enhance interactions of myeloid cells with other cell lineages. These findings propose the potential utility of CH as a predictive marker for classifying NSCLC patients. Future investigations involving ICI-treated patients and their clinical outcomes may shed light on the impact of CH, offering a basis for the development of adjuvant anticancer strategies aimed at modulating CH.

## Methods
### Patients
Description of the discovery cohort is previously reported (*Kim et al., 2023*). Briefly, 100 patients (50 LUAD, 43 LUSC, and 7 others) with metastatic NSCLC stage IV who were treated with anti-PD(L)1 (e.g., Atezolizumab, Nivolumab, and Pembrolizumab) were recruited at Samsung Medical Center (SMC) under the permission of SMC Institutional Review Board (No. 2018-04-048, 2022-01-094). All subjects provided their written informed consent to participate in the study. Whole blood sample was acquired before and 1–3 weeks after the treatment and used for genome sequencing and scRNA-seq. For replication, blood samples from additional 180 lung cancer patients (125 LUAD and 55 LUSC) were used for genome analysis (*Litchfield et al., 2021*).

### Targeted sequencing and WES
We designed a targeted sequencing panel comprising 167 selected genes associated with putative drivers of CHIP at Twist Bioscience (South San Francisco, CA). For WES, we utilized the SureSelect V5 panel from Agilent (Santa Clara, CA). CHIP-targeted sequencing and WES were both conducted at Samsung Genome Institute, respectively, employing 50 and 200 ng of genomic DNA extracted from

whole blood. The targeted sequencing panel was captured and sequenced using NextSeq 2000, with paired-end sequencing performed at a read length of 150 base pairs (bps).

## Variant filtering and calling of CHIP variants

For somatic variant calling, the Sarek pipeline (version 3.2.1) implemented in Nextflow was used (*Garcia et al., 2020*). In brief, the pipeline employed GATK's standard data preprocessing procedure and Mutect2 for somatic variant calling based on a BED file (*McKenna et al., 2010*). All passed calls were collected and annotated using Snpeff (version 4.3 and db version 105) and VEP (version 108) (*Cingolani et al., 2012*; *McLaren et al., 2016*). To filter CHIP variants, a stepwise approach based on the following criteria was applied. First, variants should meet the following conditions: (1) VAF 2–35%, (2) variant depth >500× for the discovery set and >40× for the WES replication set, (3) 4 or more (2 or more for the WES replication set) variant-covering read pairs in each forward and reverse direction, (4) global allele frequency from gnomAD <1e−5, or <1e−3 if a variant is in the COSMIC database (*Tate et al., 2019*) allele count <5% in all processed samples, and (5) no homopolymer signature found in 6 bp upstream or downstream of the candidate variant. Next, we called CHIP variants based on (1) protein sequence alterations, including exonic splicing variants, and (2) manual assessment of variant calls using IGV (*Robinson et al., 2017*). Finally, we curated data from previous studies and the COSMIC database, assigning variants reported more than 10 times or matched within a predefined list as CHIP with putative drivers (called as clonal hematopoiesis of indeterminate potential with putative drivers, or CHIP-PD) (*Niroula et al., 2021*).

## Power estimation

The power of the Poisson binomial test was calculated as the probability that a variant would be detected exclusively in one of two arbitrary groups. In brief, the most prevalent CHIP variant, *DNMT3A* p.Arg882, constitutes approximately 10% of the *DNMT3A* CHIP cases. When this variant is totally absent in the one group, a minimum of 16 *DNMT3A* CHIP patients in the other group would be required to achieve a power of 0.8. Considering the current prevalence of *DNMT3A* CHIP in our cohort (8/50 CHIP samples), this suggests that a minimum of 100 CHIP samples per group is required for determining the prevalence of any specific variant.

## scRNA-seq preprocessing and analysis

The scRNA-seq dataset was generated using the 10× Genomics 5′ single-cell kit and was partially derived from a previous study (*Kim et al., 2023*). Reads were aligned to the hg38 reference using 10X CellRanger (ver. 7.0). Contaminant RNA barcodes were removed with cellbender and sample demultiplexing was performed by demuxlet (*Fleming et al., 2023*; *Kang et al., 2018*). After filtering the cell expression matrix, we used Seurat (v 4.3.0) for downstream analysis (*Hao et al., 2021*). At the sample level, we filtered out cells with mitochondrial RNA content >15%, unique molecular identifier (UMI) counts <200 or >6000, and calculated library-size corrected counts using SCTransform (*Hafemeister and Satija, 2019*). Then, we computed PCA based on the normalized SCTransform values and integrated all samples using Harmony (*Korsunsky et al., 2019*). Using the corrected PCA, we performed unsupervised clustering via Louvain algorithms and UMAP. For each cluster, we identified marker genes using the Wilcoxon rank-sum test with presto and validated annotations using Azimuth (*Hao et al., 2021*).

## Gene set enrichment analysis

GSEA was employed to compare the expression patterns according to CHIP status based on their concordance to the hallmark pathways in the Molecular Signatures Database (MSigDB) (*Liberzon et al., 2015*). For GSEA of scRNA-seq data, we used fgsea (*Sergushichev, 2016*). Briefly, the AUC rank of DEGs obtained with and without CHIP for each cluster was used as input. Significant gene set enrichment was determined only for adjusted p < 0.05.

## scRNA-seq WGCNA

We utilized WGCNA to construct a gene co-expression network from the gene expression profiles of myeloid cells utilizing the R package hdWGCNA (*Morabito et al., 2023*). In brief, the co-expression network was constructed using a soft power of 12 for SCTransform-normalized cell counts. Within

the constructed co-expression network, we explored the type and specificity of genes in each module, excluding unassigned genes. The functionality of the black module was validated through GSEA and by examining its overlap with DEGs identified between CHIP groups (*Morabito et al., 2023*).

## GRN inference

To further understand the altered gene modulation in major cell clusters, including B cells, NK cells, CD4 T cells, CD8 T cells, and monocytes, we constructed a GRN using pySCENIC (*Van de Sande et al., 2020*). We created meta-cells for each sample using SEACells and employed these meta-cells for GRN calculation to address the sparse nature of scRNA expression and enhance resolution (*Bravo González-Blas et al., 2023*; *Persad et al., 2023*). The full transcriptome matrix and the GRN derived from meta-cells were then used to analyze regulon signatures for each cell. Finally, we mapped the regulon AUC values of each cell onto Seurat's UMAP and visualized the results using Seurat.

## Cell-to-cell communication analysis

To understand the interactions between myeloid and other cell types, we utilized the Python package CellphoneDB (*Efremova et al., 2020*), which curates human cell–cell interactions. The counts matrix, processed with SCTransform, was divided according to CHIP status and used as input for CellphoneDB. Significant interactions were then counted to determine the number of cell–cell interactions, and myeloid-related interactions were compared between samples of different CHIP status.

## Statistical analysis

We present raw data if applicable. For scRNA-seq data, scaled and log-normalized raw counts from cells are displayed in violin plots. To determine statistical significance between high-burden, CHIP, and negative groups, the Wilcoxon test or Kruskal–Wallis test was used for continuous variables and the Poisson binomial test or logistic regression for categorical variables. Statistical metrics from GESA were calculated using the fgsea package in R. All statistical analyses were conducted in R 4.2.1.

# Acknowledgements

We thank George Vassiliou for critical comments. The authors also acknowledge the Korea Research Environment Open Network (KREONET) service and the usage of the Global Science Experimental Data Hub Center (GSDC) provided by Korea Institute of Science and Technology Information (KISTI). This work was supported in part by the grants from the Korean Research Foundation (NRF-2021R1A2C3014067, NRF-RS-2023-00207857 to M Choi, and NRF-2020 R1A2C3006535 to S-H Lee), the Korea Health Technology R&D Project through the Korea Health Industry Development Institute (KHIDI) (HR20C0025 to S-H Lee), and Future Medicine 20*30 Project of the Samsung Medical Center (SMX1230041 to S-H Lee).

# Additional information

## Competing interests

Murim Choi: Reviewing editor, *eLife*. The other authors declare that no competing interests exist.

## Funding

| Funder | Grant reference number | Author |
| --- | --- | --- |
| National Research Foundation of Korea | 2021R1A2C3014067 | Murim Choi |
| National Research Foundation of Korea | RS-2023-00207857 | Murim Choi |
| National Research Foundation of Korea | 2020R1A2C3006535 | Se-Hoon Lee |
| Korea Health Industry Development Institute | HR20C0025 | Se-Hoon Lee |

| Funder | Grant reference number | Author |
|---|---|---|
| Samsung Medical Center, Sungkyunkwan University | SMX1230041 | Se-Hoon Lee |

The funders had no role in study design, data collection, and interpretation, or the decision to submit the work for publication.

## Author contributions

Hyungtai Sim, Data curation, Formal analysis, Validation, Investigation, Writing – original draft; Hyun Jung Park, Conceptualization, Data curation, Formal analysis, Investigation, Methodology; Geun-Ho Park, Yeon Jeong Kim, Data curation, Formal analysis, Investigation, Methodology; Woong-Yang Park, Conceptualization, Investigation, Methodology; Se-Hoon Lee, Murim Choi, Conceptualization, Investigation, Writing – original draft

## Author ORCIDs

Hyungtai Sim ⓘ https://orcid.org/0000-0002-3719-4190
Murim Choi ⓘ https://orcid.org/0000-0002-9195-1455

## Ethics

Patients with metastatic NSCLC stage IV who were treated with anti-PD(L)1 were recruited at Samsung Medical Center (SMC) under the permission of SMC Institutional Review Board (No. 2018-04-048, 2022-01-094). All subjects provided their written informed consent to participate in the study before taking part.

Reviewer #1 (Public review): https://doi.org/10.7554/eLife.96951.3.sa1
Author response https://doi.org/10.7554/eLife.96951.3.sa2

# Additional files

## Supplementary files

• Supplementary file 1. Supplementary tables provided in an excel file. (**a**) Clinical information of non-small cell lung cancer (NSCLC) cohort with immune checkpoint inhibitor (ICI) treatment. (**b**) Information of control samples. (**c**) List of clonal hematopoiesis of indeterminate potential (CHIP) variants detected from targeted sequencing. (**d**) List of CHIP variants detected from whole exome sequencing (WES). (**e**) Identified DEGs from annotated clusters of single-cell RNA sequencing (scRNA-seq). (**f**) Gene set enrichment analysis (GSEA) results related to myeloid lineage, related to *Figure 3C*. (**g**) Identified DEGs from each cluster by variant allele frequency (VAF) bin. (**h**) Gene-module annotation from weighted correlation network analysis (WGCNA).

• MDAR checklist

## Data availability

Due to ongoing research involving overlapping samples and privacy concerns related to individual human genotype data, we cannot make the raw genomic data publicly available. The expression data in a matrix format and metadata for scRNA-seq is deposited at K-BDS. All other derived data supporting the findings of this study are included in manuscript, supporting files or are available at: GitHub, (copy archived at *Sim, 2024*).

The following dataset was generated:

| Author(s) | Year | Dataset title | Dataset URL | Database and Identifier |
|---|---|---|---|---|
| Sim H, Lee S-H, Choi M | 2024 | Increased inflammatory signature in myeloid cells of non-small cell lung cancer patients with high clonal hematopoiesis burden | https://kbds.re.kr/BioProject/browse/view/2679124 | K-BDS BioProject, KAP241028 |

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
