## [Editor Report · eLife Assessment]

This **valuable** article represents a significant body of work that addresses some novel aspects of the biology of lung cancer, the overall influence of CHIP and its impacts on responses to therapy. While a high clonal hematopoiesis (CHIP) burden was previously linked with an inflammatory phenotype in other disease settings, the authors demonstrate with **solid** evidence that this is also true for lung cancer. CHIP is complex and more data will be required to substantiate more evidence with regard perhaps to specific mutations in certain situations and how this might influence therapy choices.

---

## [Referee Report · Reviewer #1 (Public review)]

Summary:

The study investigates the impact of Clonal Hematopoiesis of Indeterminate Potential (CHIP) on Immune Checkpoint Inhibitor (ICI) therapy outcomes in NSCLC patients, analyzing blood samples from 100 patients pre- and post-ICI therapy for CHIP, and conducting single-cell RNA sequencing (scRNA-seq) of PBMCs in 63 samples, with validation in 180 more patients through whole exome sequencing. Findings show no significant CHIP influence on ICI response, but a higher CHIP prevalence in NSCLC compared to controls and a notable CHIP burden in squamous cell carcinoma. Severely affected CHIP groups showed NF-kB pathway gene enrichment in myeloid clusters.

Strengths:

The study is commendable for analyzing a significant cohort of 100 patients for CHIP and utilizing scRNA-seq on 63 samples, showcasing the use of cutting-edge technology.

The study tackles the vital clinical question of predicting ICI therapy outcomes in NSCLC.

Weaknesses:

The study groups, comprising NSCLC patients and healthy controls, exhibit notable differences in sex distribution and smoking status. Given that smoking is a well-established factor influencing CHIP status, this introduces potential confounding variables that may impact the study's conclusions. The authors have appropriately acknowledged these disparities and provided a transparent discussion of their implications.

Comments on revised submission:

The authors thoroughly addressed all my concerns. Thank you very much for your additional work.

---

## [Author Response]

The following is the authors’ response to the original reviews.

**Public Reviews:**

**Reviewer #1 (Public Review):**
Summary:The study investigates the impact of Clonal Hematopoiesis of Indeterminate Potential (CHIP) on Immune Checkpoint Inhibitor (ICI) therapy outcomes in NSCLC patients, analyzing blood samples from 100 patients pre- and post-ICI therapy for CHIP, and conducting single-cell RNA sequencing (scRNA-seq) of PBMCs in 63 samples, with validation in 180 more patients through whole exome sequencing. Findings show no significant CHIP influence on ICI response, but a higher CHIP prevalence in NSCLC compared to controls, and a notable CHIP burden in squamous cell carcinoma. Severely affected CHIP groups showed NF-kB pathway gene enrichment in myeloid clusters.Strengths:The study is commendable for analyzing a significant cohort of 100 patients for CHIP and utilizing scRNA-seq on 63 samples, showcasing the use of cutting-edge technology. The study tackles the vital clinical question of predicting ICI therapy outcomes in NSCLC.Weaknesses:The manuscript's comparison of CHIP prevalence between NSCLC patients and healthy controls could be strengthened by providing more detailed information on the control group. Specifically, details such as sex, smot king status, and comorbidities are needed to ensure the differences in CHIP are attributable to lung cancer rather than other factors. Including these details, along with a comparative analysis of demographics and comorbidities between both groups and clarifying how the control group was selected, would enhance the study's credibility and conclusions.
**Reviewer #2 (Public Review):**
Summary:The authors used a large cohort of patients with metastatic lung cancer pre- and 1-3 weeks post-immunotherapy. The goal was to investigate whether immunotherapy results in changes in CHIP clones (using targeted sequencing and whole exome sequencing) as well as to investigate whether patients with CHIP changed their response to immunotherapy (single-cell RNA sequencing).Strengths:This represents a large cohort of patients, and comprehensive assays - including targeted sequencing, whole exome sequencing, and single-cell RNA sequencing.Weaknesses:Findings are not necessarily unexpected. With regards to clonal dynamics, it would be very unlikely to see any changes within a few weeks' time frame. Longer follow-up to assess clonal dynamics would realistically be necessary.

We truly appreciate constructive comments by the reviewers and editors. We agree with these comments and did our best to address them to improve the paper. Please see the following pages.

**Reviewer #1 (Recommendations For The Authors):**
Comment 1-1. In Figure 3B, the changes in frequency are challenging to discern. Consider employing connected line plots or another visual representation to enhance clarity and interpretation.

Thank you for the suggestion. We modified Figure 3B to efficiently visualize the changes in cell proportion. Please note that the proportional changes in cell populations were not statistically significant by treatment, pathology, or clonal hematopoiesis (CH) severity.

Comment 1-2. On page 13, Figure 3D is mentioned before Figure 3C. Please re-order to follow the correct sequence.

We corrected the sequence of the figure and revised the text accordingly.

Comment 1-3. Supplementary Figure 9 reveals an intriguing observation: the hypoxia and TNF signaling pathways appear to be regulated in opposite directions between CHIP-negative subjects and those with a Variant Allele Frequency (VAF) greater than 0.1. It would be insightful if the authors could delve into the potential implications or interpretations of this finding.

We appreciate the reviewer's insightful comment. In the GSEA results presented in Supplementary Figure 9 and Figure 3C, we specifically focused on TNF signaling in monocytes and cDCs. Our subsequent analysis revealed that the adaptation of inflammatory signals is enriched in the myeloid cells in the CHIP-severe patients (Supplementary Fig. S12). Following the reviewer’s comment, we found that the leading-edge genes were shared between the TNF signaling and hypoxia pathways in most clusters (Supplementary Fig. S15). Suggested core genes, such as *FOS*, *DUSP1*, *JUN*, and *PPP1R15A,* play critical roles in the inflammatory phenotypes of myeloid lineages. Based on this finding, we added a paragraph in the Discussion section to address the implications of these shared signatures as follows (lines 340-348):

“Our GSEA results specifically indicated the enrichment of TNF signaling and hypoxia pathways in most clusters of patients with severe CH (Supplementary Fig. S9). The leading-edge genes from GSEA results showed core genes such as FOS, DUSP1, JUN, and PPP1R15A, which are known to play critical roles in the inflammatory phenotypes of immune cells, were shared between the TNF signaling and hypoxia pathways in all significant clusters. (Supplementary Fig. S15). Furthermore, gene regulatory network analysis using SCENIC indicated a higher enrichment of inflammatory signatures in myeloid lineages (Supplementary Fig. S9), highlighting the pronounced inflammatory phenotype of CH clones in these cell lineages.”

Comment 1-4. The plots in Supplementary Figure 12 would benefit from enlargement to improve legibility and facilitate a better understanding of the data presented.

We improved resolutions and enlarged Supplementary Figure S12.

**Reviewer #2 (Recommendations For The Authors):**
Comment 2-1. The authors state that CHIP is seen at a higher prevalence in the metastatic lung (44/100) vs controls (5/42), however, no in-depth info other than age is given about the normal cohort (Table S2). It would be important to make sure the cohorts are matched with regards to smoking hx, age range, etc before making the claim that CHIP is more frequent in the metastatic lung cancer group.

Thank you for the comment. To provide additional information of control cohort including current smoking habits and their sex information, we added columns in Table S2. While we tried to match the age distributions between the control group without a history of solid cancer and the lung cancer cohort, we observed that the lung cancer cohort had slightly older ages (mean ages: 58.9 vs. 64.1 years), a higher prevalence of smoking (current smokers: 11/42 vs. 37/100), and a higher proportion of males (male/female: 18/24 vs. 91/9). Age and smoking are well-known epidemiological contributors to lung cancer and could influence the prevalence of clonal hematopoiesis (CH).

However, previous studies have reported similar prevalence rates of CH in NSCLC patients, which aligned with our findings (Bolten et al., 2020 Nat Genet; Hong et al., 2022 Cancer Res). Moreover, our most prevalent CH mutations, including *DNMT3A*, *TET2*, and *PPM1D*, were marginally affected by smoking, and this trend has been consistently observed in healthy populations (Levin et al., 2022 Sci Rep). We have acknowledged these factors as major limitations of our study in the Discussion section as follows (lines 379-390):

“Also, the distinct characteristics of our cohort can be confounders for our results. Compared to control patients, our cohort was biased toward slightly older ages, higher prevalence of smoking habits, and with a higher proportion of males (mean age: 64.1 vs. 58.9; current smokers: 37/100 vs. 11/42; male/female: 91/9 vs. 18/24 Supplementary Figures S1 and S3). However, previous studies have reported similar prevalence rates of clonal hematopoiesis in NSCLC patients, aligned with our findings (9,51). Moreover, our most prevalent CH mutations, including DNMT3A, TET2, and PPM1D, were marginally affected by smoking, and this trend has been consistently observed in both healthy populations and NSCLC patients (10,51,52).”

Comment 2-2. Figure 1 - 1A states there were 100 CHIP and CHIP-PD mutations identified, but in 1B, C, and D there are < 100 bars and/or dots shown. How were the mutations in 1A then triaged to be shown in 1B-D?

It appears that our poor annotation caused this misunderstanding. In Figure 1A, we showed the number of samples in each study group but did not provide detailed information in the legend. We found 67 mutations among the 100 patients and presented the mutational statistics in Figures 1B–D. Accordingly, we have revised the Figure 1 legend to clarify this sentence “The numbers indicate sample counts in each group.” (lines 426-427).

Comment 2-3. Table S4 - would be helpful to have # of variant reads and # of total reads as columns (and also calculate VAF for an additional column).

Thank you for the comment. We added columns revealing the total number of reads and the number of variant reads in Table S4. Also, we calculated the VAF and included it as a new column as suggested by the reviewer.